# Story to Change Culture on Early Childhood in Australia

**DOI:** 10.3390/children10020310

**Published:** 2023-02-06

**Authors:** Nathaniel Kendall-Taylor, Annette Michaux, Donna Cross, Karen Forde

**Affiliations:** 1FrameWorks Institute, Washington, DC 20005, USA; 2Parenting Research Centre, East Melbourne, VIC 3002, Australia; 3Telethon Kids Institute, 15 Hospital Avenue, Nedlands, WA 6009, Australia

**Keywords:** early childhood, public policy, issue framing, cultural models, narrative change

## Abstract

The goal of the article is to support the early childhood sector’s efforts to increase the salience of early childhood as a social issue and change policy and practice to better support young children and their families. Cultural models shape how people think about social issues and support solutions. Changing how issues are framed—how they are presented, positioned and focused—can help shift these models and facilitate culture change. Using mixed methods research, we identified cultural models that members of the Australian public use to think about early childhood and compared these mindsets to concepts that the sector seeks to advance. This revealed a set of gaps in understanding that make it difficult for the sector to advance its agenda. We then designed and tested framing strategies to address these challenges and improve the salience of early childhood as a social issue, increase understanding of key concepts and build support for policies, programs and interventions. Findings point to strategies that advocates, service providers and funders can use to communicate more effectively about the importance of the early years.

## 1. Introduction

Issue framing—choices in how information is presented and positioned—has been an important part of creating social change [1]. On issues such as tobacco control [2], mental health [3]), and marriage equality [4], conducting research to understand how people think and how this thinking can be shifted with different ways of positioning issues has been a vital part of changing public policy.

In 2017, the Telethon Kids Institute and the FrameWorks Institute began a three-year project to study Australian cultural models of early childhood and test frames for their ability to move these mindsets. Our goal was to support the early childhood sector’s efforts to increase the salience (relative importance) of early childhood as a social issue and change policy and practice to better support young children and their families.

The project builds on other efforts, particularly those in the United States led by the Center on the Developing Child at Harvard University [5] and in Canada by the Alberta Family Wellness Initiative [6] to apply the science of framing [7] to improve public and policy maker understanding and use of the science of early childhood development [8]. The current research and these other existing efforts are rooted in communicating three core ideas that have emerged through decades of research by a range of developmental scientists working across disciplines. First, that the architecture of the developing brain is built through a bottom-up process whereby a young child’s experiences interact with genetic predispositions to shape biological systems and life-long outcomes [9]. Second, that early experiences of chronic and unbuffered adversity can create a toxic stress response that derails the process of healthy development [10]. Finally, that supportive and responsive relationships with nurturing caregivers are key in assuring healthy development and buffering against the detrimental of effect of chronic adversity [11]. While the science of development is constantly developing—for example, with a new focus on biological systems other than the brain (the microbiome for example), with outcomes other than learning (life-long health for example), and with structural factors other than relationships (structural racism, for example) as key drivers of developmental processes—the project described below is grounded in these three core tenets of the science of development and driven by the goal of better communicating them outside the early childhood sector. 

The project aimed to answer two questions: (1)How do messages that the early childhood sector is seeking to communicate compare to cultural models that the Australian public uses to think about this developmental period?(2)How can the sector frame messages to increase the salience of early childhood and improve public demand for policies and practices that support early development?

The project is rooted in two areas of social science theory. Drawing from research on frame effects, we begin from the assumption that understanding is frame-dependent [12]. That is, patterns in the way that information is presented affect how people think about, value and behave on social issues [13,14]. The effects of framing on public opinion stem from the psychological process of ‘accessibility’ [1]. This is the idea that the frames in a message cue particular mental representations, which become active and accessible as people make sense of information, formulate opinions and make judgements [15,16]. The literature on frame effects focuses on how variations in issue presentation affect outcomes related to understanding, emotion and behavior [1].

Second, we use cultural models theory from psychological anthropology [17], which advances the idea that, in addition to its external aspects in social institutions and the material world, culture exists in the mind and is shared by members of a social group [18,19,20]. From this view, culture is a set of shared, implicit mental models that individuals use to make sense of information, experiences and their social worlds; process information; interact with others; and make decisions.

Understanding the cultural models that people implicitly draw on to make sense of issues provides communicators with information on how their messages will be understood and the ability to predict message effects. Studying cultural models also helps communicators see and focus on aspects of an issue where current patterns of thinking create communication challenges and where presenting information in different ways might facilitate different understandings and ways of supporting policies, programs and interventions. Finally, a cultural models analysis assists in formulating hypotheses about effective ways of reframing issues and messages that can be empirically tested. 

## 2. Materials and Methods

The project built on others conducted in Australia over the last 10 years on reframing parenting [21], reframing brain development, and early education [22].

The project consisted of two phases: descriptive and prescriptive research. Following a detailed description of the research project, its protocols, purpose, and outcomes, written and, where possible in qualitative research, verbal consent was sought from and provided by all participants [23]. Below, we provide an overview of the methods used in these two phases. 

### 2.1. Descriptive Research

To identify concepts from the early childhood sector that need to be communicated and the cultural models that members of the public use to make sense of these concepts, we used expert and cultural models interviews (see appendices for questions). 

For expert interviews (see Appendix A for interview protocol), we employed an open-ended guide to elicit participant views on the key dimensions of early childhood development that need to be translated [24]. Fifteen interviews were conducted in 2018 and recorded and transcribed for analysis. The sample of interviewees was purposefully selected to represent expert voices across disciplines and fields working on early childhood-related issues (e.g., psychology, human development, education, public health and neuroscience). Participants were asked a series of open-ended questions designed to elicit their thoughts on the most important early childhood concepts for members of the public to understand, the key policies they thought were important to increase support for, and their aspirations for what a society that more robustly supported young children and their families would look like. 

Interviews were analyzed using a grounded theory approach [25]. Following this analysis, three separate groups of 8 to 12 individuals from the sector received a draft of the key points of consensus emerging from the analysis and participated in a researcher-moderated session. The sessions were facilitated to focus discussion on three broad questions: what concepts were missing from the key points, what concepts were included but were less than essential and could be winnowed, and which included concepts were important but required additional refinement. 

Cultural models interviews (see Appendix B for interview protocol )—a method from psychological anthropology—were used to elicit explanations, examples and stories from members of the public in Australia about early childhood [26]. In 2018, 26 participants were purposefully recruited to represent variation in race/ethnicity, gender, age, residential location, educational background, political affiliation and family structure.

We analyzed data from these interviews, using both grounded theory and cultural models analysis [26,27]. Three researchers independently analyzed the data to identify social discourses, or common, standardized ways of talking, across the sample using a grounded theory approach that emphasized constant comparison between emerging patterns and transcripts [25]. These social discourse themes were then independently analyzed by the same three researchers for tacit organizational assumptions, propositions, and connections that were commonly made but taken for granted throughout an individual’s transcript and across the sample [28].

Findings from expert and cultural models interviews were then brought together in a comparative analysis that looked at overlaps and gaps [29].

### 2.2. Prescriptive Research 

Working from a set of gaps, we then developed a set of more than 25 frames and framing hypotheses to test. Among other things, these frames included values [30,31], issue frames [32] and explanatory metaphors [33,34]. 

After developing a list of framing hypotheses, we began a three-part testing process. First, to test the utility of particular frames, we conducted 76 on-the-street interviews. Recruited in public places, participants were asked to talk about an issue unaided by a frame. They were then introduced orally to a single frame (a value or a metaphor, for example). We then asked additional questions to provoke further discussion about issues related to early childhood. Video data were collected with participants’ consent and analyzed to see whether and how various frames affected people’s thinking and use in conversation. 

Twelve frames (values and metaphors, for example) and framing strategies (for example, temporal ways of emphasizing the positive outcomes of supporting early childhood development) showing promise in improving understanding of core concepts and support for policies and programs were advanced to a series of nationally representative experimental surveys, to quantitatively test the relative efficacy of different framing strategies [35,36]. Survey experiments were web-based and used random assignment to expose 7265 participants to either a control or one of a set of framed messages built around values, metaphors or other frame elements. Each participant was then asked a series of questions to assess understanding of target concepts, issue salience, attitudes and support for a variety of early childhood programs and policies. 

The most effective frames from the experimental surveys then underwent a second round of qualitative testing with 54 participants in focus groups to evaluate how well a given frame held up in social interactions as it was used and shared. 

At the end of this empirical testing process, we had a co-designed set of evidence-based framing recommendations that addressed gaps between sector and public understandings of early childhood development. 

## 3. Results

### 3.1. Descriptive Research Results 

Our analysis revealed points of overlap between sector and public understandings of early development. These overlaps represent common ground that communicators can build on to translate key ideas, build understanding and increase support for policies, programs and interventions [5,8,37]. The sector can amplify these ideas knowing that there is general agreement on these concepts. 

Members of the early years sector and the public share the following understandings:Early childhood is a time of significant and rapid developmentDevelopment involves the brainNutrition is a key determinant of developmentSkill acquisition is an important part of development (e.g., social, emotional and problem-solving skills)Financial resources shape a child’s opportunities in ways that can affect developmentThe adults a child interacts with shape developmental outcomesPlay is a basic human instinct and is essential for positive learning and development

In addition to these overlaps, we found a set of gaps between sector and public understandings. The following areas could be addressed to improve public understanding of early childhood, elevate the salience of the issue of early childhood, and build support for policies and programs [29].
**1.** **Early Childhood as a Key Window of Development: One of Several vs. The One and Only.**

According to the sector, the pre- and peri-natal periods and first five years of life are foundational times of development [38]. However, participants emphasized that these are not the only times when development is open. Adolescence is another key developmental period, and they stressed that while pronounced in these sensitive periods, plasticity exists over the life-course [39]. For the public the early years are characterized by an almost infinite ability to absorb knowledge and skills, but the public also assumes this window of opportunity closes sometime between the ages of five and seven.
**2.** **Systems and Outcomes: Interconnected and Wholistic vs. Brains and Learning.**

The sector explained that biological systems throughout the body shape and are shaped by one another, that they function as an integrated whole, and that this integration and its importance is central to a wholistic concept of development [40]. While the public recognize the importance of brain development and its connection to learning, the connections between the brain and other systems are not understood. As a result, there is an underappreciation of other biological systems and a lack of focus on health as an outcome of development.
**3.** **Early Mental Health: Complex and Fundamental vs. Simple or Non-Existent.**

Sector participants explained that the foundations for children’s mental and emotional health are laid early, and that children can feel intense emotions and be deeply affected by them in infancy [41]. The public, however, assumes that children only start being affected by mental states, emotions and stress around the ages of 5 or 6 and, thus, do not focus on or feel motivated to address child mental health in the early years.
**4.** **Effects of Adversity: Significant but Reversible vs. No Harm or ‘Damage Done Is Damage Done’.**

Sector members argued that early adversity can have substantial and durable effects on developmental outcomes. However, with the right supports, children experiencing adversity can achieve positive outcomes [42]. The public, by contrast, toggles between thinking that adversity does not impact young children because of their lack of language and memory and their general resilience and thinking that adversity makes an indelible imprint on children, dooming them to negative lifelong outcomes.
**5.** **Environments: Structural Factors, Environments and Relationships vs. People and Money.**

Sector members see early development as shaped by many socioeconomic and structural factors, as well as relationships and interactions [43]. While the public thinks that ‘environments’ influence children’s development in the early years, ‘environments’ is primarily taken to mean the adults in children’s lives. The public has a less systemic, narrower understanding of the factors shaping development.
**6.** **Responsibility: Society First vs. Individuals First.**

According to sector participants, Australians have a collective responsibility to support positive early development for all children. They advocate for policies and interventions that would change the socioeconomic environment and reduce social inequalities, create a physical environment that supports development, and improve social systems’ capacity for early intervention [44]. While the public agrees that government has a role to play, there is not a concrete understanding of what that role should entail. Instead, the public assumes that responsibility lies with parents and caregivers.
**7.** **Who Needs Better Support? Everyone According to Their Needs vs. ‘Those People’.**

The sector argues for developing and implementing a strategic effort to reach underserved children and families, alongside a broader effort to provide inclusive environments and services for all families [45]. The public, on the other hand, reasons that help should focus only on underserved families and communities, because of a deeply seated belief that poor developmental outcomes are ‘those’ people’s problems.

### 3.2. Prescriptive Research Results 

Over the past decade, the early childhood sector in Australia has come together around an effort to more effectively translate the science of early development for audiences outside of the sector. This is evident in the work of groups such as Centre on Community Child Health, the Parenting Research Centre, ARACY, and others. This work has been successful in moving public thinking and convincing people that early childhood matters. Despite this success, these efforts have not sufficiently elevated the issue on the public policy agenda. People agree that ‘early matters’, but they do not put young children first when it comes to resource allocation [46]. 

Successes to date provide a springboard to move the early childhood agenda further and more broadly. Our findings point to a new narrative strategy that can help make this work even more effective. This does not mean the sector should stop explaining the science of development as it has successfully been doing, rather that this science explanation needs to sit in a broader narrative.

The following five strategies emerged from prescriptive research as key areas for early childhood communicators to consider for effectively moving the early years up the social and policy agenda. Together they constitute a new narrative about the importance of the early years and what society must do to support this sensitive period of development.
**1.** **Make early development and learning about improving children’s health and wellbeing.**

Connecting early childhood development and learning to children’s health expands people’s thinking and increases the salience of early childhood, moving it up people’s list of priorities and inspiring more support for solutions. People recognize that early childhood is important, but they do not see childhood as a social issue or think that more can be done beyond educating parents to take their responsibilities more seriously. 

Talking about early development and learning as ways to support children’s health helps people see early development in a new light and raises the profile of the issue. People recognize health as an important issue that needs to be at the top of the country’s political and social agenda.
**2.** **Talk about effects on health and wellbeing in the present and future.**

Making the connection between now and later helps people see that supporting early childhood development leads to good health and wellbeing for kids and families now, as well as for individuals and society in the future. 

People tend to prefer smaller, immediate rewards to more significant future ones [47]. This affects how we make decisions and the policies we support. Highlighting the immediate benefits of supporting early development addresses our tendency to undervalue future benefits. However, keeping the future in messages maintains the ability to highlight the true social benefits of supporting early development and the repercussions of not supporting children and families.
**3.** **Define the problem as the way a lack of support is leading to poor health for some children.**

Our research showed that highlighting that some children do not have what they need to develop well and that this is leading to disparities in children’s health was highly effective. This dimension of the story raises the profile of early childhood development as an issue warranting public attention and drives support for policy change.

Stating that some children in Australia lack good health and connecting these differences to uneven support helps people recognize there is a problem needing to be addressed. This overrides people’s assumption that Australia is already doing enough to support children and families and that not much else can or should be done [48]. It also overcomes the idea that positive early development naturally happens and clarifies that positive development requires support. 

Highlighting disparities in the support children receive and, in turn, their health, activates Australians’ strong sense of fairness and justice when it comes to children.
**4.** **Frame the call to action with an explicit appeal to fairness and the idea that we need to ensure that children can thrive, no matter where they live.**

Appeals to the public’s strong sense that society should be fair and equal—especially for children—are compelling and raise a productive sense of urgency to address early childhood issues. In making a case for action, the appeal to fairness establishes why the issue of early childhood development matters, and why people should care about it and engage to help. It moves people to action and boosts support for change.

Appealing to fairness taps into a widely shared Australian value [21,49]. People know that things such as quality early childhood education and care are unaffordable for families on low incomes. They agree this is not fair or right. A focus on fairness across places gets people thinking about differences in contexts and circumstances and how these differences affect children. It does this without triggering blame, negative stereotypes, or ‘us vs. them’ thinking.
**5.** **Talk about supporting every child and community according to their needs.**

Emphasizing the importance of helping everyone according to their needs and the needs of their communities builds support for targeted responses and actions without stigmatizing communities. By arguing for support for every child and community according to their needs, we avoid ‘othering’ certain communities and setting up a zero-sum mentality about public resources in which more for some groups means less for others, which has been found to decrease support for equity issues [48,50].

Talking about the need to help ‘every’ child, family, and community wards off debates over who deserves what and the reaction that more for ‘them’ means less for ‘me’—a response that surfaces when messages focus only on helping specific communities. The universal dimension of the message also avoids activating the assumption that differences in outcomes are the result of ‘those’ people’s or communities’ supposed lack of morals or poor decisions and habits [46]. 

Together these recommendations form a narrative strategy (see Figure 1: A new narrative strategy for the early childhood sector in Australia). 

## 4. Discussion

These findings provide a new strategy that the early childhood sector can use to increase the salience of early development as a social issue, improve understanding of key dynamics of this issue, and build support for solutions. This type of work has been used to good effect in other parts of the world [6,7]. 

Some of these recommendations are at odds with current practice—for example, a strong focus on future instrumental benefits of development (the return on investment argument [51] or framing around vulnerability and risk rather than a more targeted universalist presentation and approach. Most significantly, these findings recommend broadening the issue frame from early learning to a focus on early and lifelong health and learning. This is likely to be challenging in the Australian sector where the early development has long been framed around early learning, as can be seen, for example, in Australia’s National Quality Framework [52]. 

In addition to this narrative strategy, having a better understanding of where sector and public understandings overlap and diverge provides a way of understanding the success and failure of other strategies. This type of work has, for example, been of utility for the ageism field in the United States in its thinking about the effects of image use [53]. The alignment between sector and public understandings of the importance of the early years as a critical developmental window and the benefits of public investment has been used in efforts over the past 10 years—such as the Every Benefits and Thrive By Five campaigns—to make childcare a central policy issue [54,55]. It was a key election platform for both major parties during the 2022 federal election campaign. On the other hand, although we are seeing some positive change, efforts from advocates to prevent child abuse and neglect have tended to focus on vulnerability and disadvantage [56], reinforcing the public belief that individual parenting deficits in ‘those’ families mean little can be done in public policy, as the fault lies with parents [21]. This framing gap has hindered understanding of the broader determinants of child abuse and neglect and led to limited policy change [29].

Understanding these overlaps and gaps can help communicators make better decisions about how to position messages and craft organizational strategy. This also allows a clear view of the ways that key sector ideas are either at odds with or not yet fully grasped by the public. This helps communicators recognize that they are not the audience of their messages and adapt their framing to these gaps and overlaps, which lies at the heart of effective communications.

There are a number of limitations associated with this work. Three are particularly important to note here. First, while we are confident that the mindsets identified here are broadly held across the Australian population, there are certainly variations across groups and between individuals in the degree to which these mental models are held and their role and relative dominance in decision making. This research was not designed to locate those differences, but subsequent research could measure the distribution and strength of these mindsets across demographic groups in Australia. These differences are key to understand in most effectively applying the recommendations detailed here. Second, this research did not test the instantiation of the strategies outlined here in actual messages and communications. Subsequent research should use A/B testing to apply the framing recommendations laid out here in real world communications materials, systematically expose groups to messages that vary in their use of the strategies, and measure relevant outcomes including issue salience and policy support. Finally, while the framing strategies presented here address critical gaps in understanding between the early childhood sector in Australia and members of the general public, they most certainly not exhaustive. In other words, our recommendations are limited by the ideas that were tested and there are likely other strategies that should be empirically explored for their ability to bridge gaps in understanding and improve understanding and use of the science of early childhood in Australia. 

## 5. Conclusions

Our project illustrates the importance of an empirical approach to science translation and social issue communication. The same methods used to study early childhood development and support can be used to understand how people think about these issues and what moves them to demand action and new solutions. 

We have learned that simply doing communications research and looking at questions of how best to position an issue does not ensure uptake of the resulting recommendations. Like any knowledge creation project, impact requires a focus on implementation. To address this, we created tools to support the use of this research—including an online self-guided course, a toolkit, a strategic brief and a framing guide [57]. However, products alone do not translate into knowledge use or policy and practice change. Building this new story into the sector’s communication channels will require partnership with key organizations as they adapt the narrative to communicate their issues (child mental health, for example) through particular channels (social media, for example), and collaboration between organizations to support and reinforce application. Public thinking will not be changed overnight. There is a need for a new long-term strategy to see the shifts in the mindsets necessary to change the policy context for young children and families. 

## Figures and Tables

**Figure 1 children-10-00310-f001:**
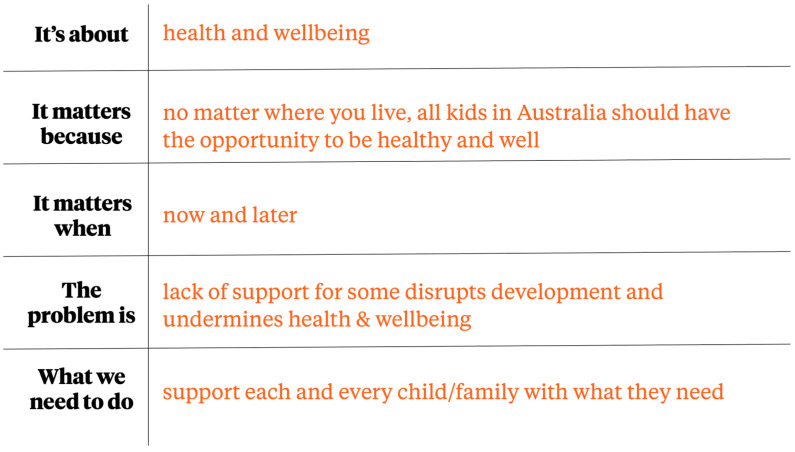
A new narrative strategy for the early childhood sector in Australia.

## Data Availability

Data is available upon request from corresponding author.

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
