# Peer review of "Story to Change Culture on Early Childhood in Australia"

_children, 2023, doi:10.3390/children10020310_

Round 1

Reviewer 1 Report

The topic and the research work are interesting and is focused on changing culture on early childhood (in Australia).

Comments and suggestions

Line 18.- ‘Childhood; Public Policy, Issue Framing;…’

Lines 107-108.- ‘Participants were asked a series of open-ended questions designed to elicit their…’. Why not include these open-ended questions in an annex?

Lines 242; 330 .- To be deleted

Line 251.- ‘The following five strategies emerged from prescriptive research as key areas for early childhood communicators to…’. -> the strategies are listed in indifferent paragraphs. Perhaps including numbers or bullets would make it clearer.

Aim of the article

In lines 29-32 and 55-59, the aims of the project are written. However, the aim of the article (in the abstract and in the article) should be clearly stated.

Limitations

Are there no limitations derived from your research approach?

References

Line 445.- Australian Children’s Education Care Quality Authority. https://www.acecqa.gov.au/nqf/about ; ?; yellow? date?

Line 455.- Empty line

Line 456 & 457.- Is it located in the right place?

Author Response

Response: Reviewer 1

We wish to take the opportunity to thank the first reviewer for a very careful read of the manuscript and for the comments offered in service of improving the submission. We appreciate the reviewer’s time and attention and have made efforts to address all of the reviewer’s comments and suggested edits. We believe the paper is stronger as a result!

The following is a more detailed account of changes made in response to the reviewer’s comments:

  • Line 18.- ‘Childhood; Public Policy, Issue Framing;…’ Change made.
  • Lines 107-108.- ‘Participants were asked a series of open-ended questions designed to elicit their…’. Why not include these open-ended questions in an annex? Questions have been added in two appendices—one containing interview guides used for expert interviews and a second for cultural models interviews.
  • Lines 242; 330 .- To be deleted. Change made
  • Line 251.- ‘The following five strategies emerged from prescriptive research as key areas for early childhood communicators to…’. -> the strategies are listed in indifferent paragraphs. Perhaps including numbers or bullets would make it clearer. Done!

Aim of the article

In lines 29-32 and 55-59, the aims of the project are written. However, the aim of the article (in the abstract and in the article) should be clearly stated. We have added text to the abstract to make this clear.

Limitations

Are there no limitations derived from your research approach? There most certainly are! We have added a paragraph (line 398-414) laying out what we think are the three most important limitations of this work.

References

Line 445.- Australian Children’s Education Care Quality Authority. https://www.acecqa.gov.au/nqf/about ; ?; yellow? date? We have revised to address this.

Line 455.- Empty line We have taken out this extra space.

Line 456 & 457.- Is it located in the right place? We have moved this reference

Reviewer 2 Report

The research is quite interesting and important. 

I would like to point out that there is a lack of citations or supported research in the Results section. I would argue that there are several important ideas that might be found in other research as well. 

Other than that, perhaps a revisiting of typsetting and the article is good to go.

Author Response

Response: Reviewer 2

Thank you to the reviewer for taking the time to go through the article and provide feedback. Adding citations into the results section was an important comment and is the main change that I have made to the manuscript in response to the reviewer’s comments. Thank you very much again, for your time and attention to the article and see below for more specific notes in response to each of the reviewer’s comments.

Comments:

  1. The research is quite interesting and important. Thank you and, once, again, we appreciate the reviewer’s time on the piece and attention to how we can make improvements to the manuscript.
  2. I would like to point out that there is a lack of citations or supported research in the Results section. I would argue that there are several important ideas that might be found in other research as well. We have made numerous additions to the results section to assure that concepts and ideas discussed elsewhere are better referenced and more robust citations provided. We believe this strengthens the piece and thank the reviewer for calling this out. Changes include the following:
    1. Page 5, lines 182-183: references were added to work where expert-public overlaps have been discussed and used to generate framing tactics and translational strategies.
    2. Page 5, lines 199-200: a reference was added here to a published article that uses the idea of “gaps in understanding” to triangulate communication challenges and design and test strategies that can help to bridge them.
    3. Page 5, line 204 and page 6, line 207: references on critical/sensitive periods was added.
    4. Page 6, line 214: reference was added to a paper over-viewing the research on the connections between brains and other biological systems in the developmental process.
    5. Page 6, line 223: reference was added about early childhood mental health.
    6. Page 6, line 230: reference was added about plasticity and stability.
    7. Page 6, line 237: reference was added about structural factors affecting early childhood development.
    8. Page 6, line 246: reference was added about the need for social policy level reforms to better support early childhood.
    9. Psge 7, line 254: reference was added to article on the perspective of targeted universalism in early childhood policy.
  3. Other than that, perhaps a revisiting of typsetting and the article is good to go. I agree that the formatting is funky in places. We have made an attempt to address some of this in places but a number of these arise from using the journal’s template and thus our hope is that they will be fixed in the final mock up of the article.